# Mental Health Literacy and Attitudes Towards Mental Health Problems Among College Students, Nepal

**DOI:** 10.3390/bs14121189

**Published:** 2024-12-13

**Authors:** Dev Bandhu Poudel, Loujain Saud Sharif, Samjhana Acharya, Alaa Mahsoon, Khalid Sharif, Rebecca Wright

**Affiliations:** 1Department of Humanities and Social Sciences, G.P. Koirala Memorial Community College, Kathmandu 44602, Nepal; 2Department of Humanities and Social Sciences, Brooklyn College, Kathmandu 44600, Nepal; 3Rupy’s International School (A-Level)—Cambridge Associate School, Kathmandu 44600, Nepal; 4Faculty of Nursing, King Abdulaziz University, Jeddah 21589, Saudi Arabia; lsharif@kau.edu.sa (L.S.S.); mahsoon@kau.edu.sa (A.M.); 5Central Department of Rural Development, Tribhuvan University, Kathmandu 44600, Nepal; samjhoo44@gmail.com; 6Department of Behavioral Medicine and Psychiatry, West Virginia University, Morgantown, WV 26505, USA; khalid.sharif.md@gmail.com; 7School of Nursing, Johns Hopkins University, Baltimore, MD 21205, USA; rebecca.wright@jhu.edu

**Keywords:** attitudes, first aid skills, help-seeking, mental health literacy, mental health problems, self-help strategies, shame, stereotypes, students, Nepal

## Abstract

(1) Background: Research on mental health literacy (MHL) and attitudes toward mental health problems (ATMHP) among non-medical college students in Nepal is limited. This study examined the relationship between MHL and ATMHP, considering demographic variables and familiarity with mental health issues; (2) Methods: We conducted a cross-sectional survey with 385 college students from Chitwan and Kathmandu, Nepal, using opportunity sampling. Descriptive and inferential statistics examined demographic differences, while Pearson’s correlation assessed relationships among latent variables; (3) Results: No relationship was found between MHL and ATMHP (r = −0.01, *p* = 0.92). Females had greater awareness of stereotypes (*p* = 0.025, g = 0.24). Hotel management students showed better self-help strategies (*p* = 0.036, *d* = 0.46). Public college students scored higher in self-help strategies than government (*p* = 0.036, *d* = −0.32) and private college students (*p* = 0.02, *d* = −0.32). Non-employed students outperformed employed ones in self-help strategies (*p* = 0.002, g = −0.46). Other demographic factors showed no significant relationships; (4) Conclusions: MHL and ATMHP were unrelated, indicating that increasing MHL alone may not improve attitudes. Multidimensional interventions combining education and experiential learning are needed. Certain demographic factors influenced stereotypes and self-help strategies, while others showed no significant impact.

## 1. Introduction

Mental health literacy (MHL) and attitudes toward mental health problems are crucial components in shaping individual and societal responses to mental health issues. MHL encompasses the ability to recognize mental disorders, access relevant information, understand risk factors, engage in self-treatment, and seek professional help [1,2]. Attitudes toward mental health, including stigma and personal beliefs, significantly impact individuals’ willingness to seek help and support those with mental health issues [3,4,5].

Understanding the interplay between MHL and attitudes toward mental health problems is essential for developing effective mental health interventions. Research indicates that higher MHL is associated with better understanding and more favorable attitudes toward mental health [6,7]. However, this relationship is not always straightforward, with some studies suggesting that increased MHL does not necessarily translate into reduced stigma or negative attitudes [8,9]. This complexity underscores the need for further investigation into how MHL and attitudes interact and influence each other.

Demographic factors such as age, gender, ethnicity, and religion have been shown to influence both MHL and attitudes toward mental health. For example, younger individuals and females generally exhibit higher MHL [10,11], while gender differences in attitudes often reveal that males may hold more negative views toward mental health issues compared to females [12,13]. Ethnic and religious backgrounds also impact MHL and attitudes [14], though findings vary across studies [15]. However, in the context of Nepal, these factors are underexplored and understood. Additionally, further studies are essential to investigate the reasons for these variations.

Academic background and geographic location further complicate the relationship between MHL and attitudes. Higher education levels are generally associated with better MHL [16,17], but the impact of academic field and geographic location on MHL and attitudes is inconsistent [12,18]. For instance, some studies report no significant differences in MHL across different districts or academic levels [12,13,14,15,16,17,18,19], while others highlight regional variations in mental health knowledge and attitudes [20]. Additionally, academic advisors in teaching roles exhibit greater mental health literacy than their non-teaching counterparts, highlighting the influence of educators on mental health awareness [21].

Research on the relationship between mental health literacy (MHL) and attitudes toward mental health problems (ATMHP) has shown inconsistent findings, with some studies indicating that higher MHL improves attitudes [6,7,8,9,20,22,23,24,25], while others suggest even mental health professional had stigma towards mental health problems [26,27,28]. We found no research providing sufficient data with evidence of reliability to investigate whether mental health literacy (MHL) minimizes negative attitudes toward mental health problems across the general population in the distinct cultural context of Nepal. Additionally, there is a lack of research specifically focusing on university and college students in this area of research. Given their role as future leaders and influencers, university students are crucial in shaping societal norms. Their mental health beliefs and behaviors are particularly significant for reducing stigma and promoting mental health awareness, potentially reflecting broader societal trends.

We aim to explore the relationship between MHL and ATMHP among university and college students, addressing these inconsistencies and examining how demographic factors influence these variables. This study fills critical gaps in the literature and provides insights to guide targeted interventions and policies aimed at enhancing mental health literacy and reducing stigma in Nepal.

The theoretical framework for this study integrates the models proposed by Jorm et al. (1997) and Gilbert et al. (2007) to explore the relationship between mental health literacy (MHL) and attitudes toward mental health problems (ATMHP) [1,29], Jorm et al. (1997) define MHL as encompassing knowledge of mental health conditions, risk factors, treatment options, and self-management strategies, as well as attitudes that influence stigma and help-seeking behaviors. These attitudes are critical for understanding how beliefs shape perceptions of mental health, making them central to this study [1]. Complementing this, Gilbert et al. (2007) emphasize the role of social rank theory and shame, suggesting that cultural and societal norms reinforce internal and external shame, which significantly affect help-seeking behavior [29]. This integration highlights how MHL can improve ATMHP by fostering understanding and reducing stigma while addressing negative attitudes, which can enhance MHL by increasing openness to mental health knowledge. Given the cultural context of Nepal, where stigma and limited awareness pose barriers, examining these variables together provides a comprehensive approach to understanding and addressing mental health challenges.

## 2. Materials and Methods

### 2.1. Study Design

We employed a cross-sectional design to investigate the connection between specific demographic variables and mental health literacy (MHL), as well as attitudes towards mental health problems (ATMHP). The focus was on assessing how different demographic variables influence each of these variables independently. In addition, the study investigates the connection between MHL and ATMHP.

### 2.2. Participants and Procedure

According to Cochran’s formula, the minimum required sample size for a population with an uncertain proportion was calculated as follows: n_0_ = (Z^2^pq)/e^2^, where n_0_ is the sample size, Z is the z-score for the desired confidence level, p is the estimated proportion of the population, q is 1 − p, and e is the desired level of precision (i.e., marginal error) [30,31]. Assuming a confidence level of 95% (Z = 1.96), an estimated population proportion of 0.5 (p = 0.5, q = 0.5), and a marginal error of 5% (e = 0.05), the minimum required sample size was calculated as ((1.96)^2^ (0.5) (0.5))**/**(0.05)^2^ = 385 [30,31]. Therefore, we included 385 participants aged 18 to 24 years, 348 from Chitwan district and 37 from Kathmandu district in Nepal. Opportunity sampling was employed to recruit college/university students as participants who are crucial for understanding and influencing societal attitudes toward mental health [32,33].

The sample included various demographic factors, with gender categorized as male and female, which represented a binary classification aligned with biological sex, ethnicity, location, academic levels, fields of study, institution types, and familiarity with mental health issues. Participants were surveyed using both online platforms, Google Forms and paper-pencil methods. Online portals such as Facebook, LinkedIn, Instagram, and Gmail were utilized to access the information online, and the participants were visited on campus during the paper-based survey. The based method was utilized in three different colleges from Chitwan and one college from Kathmandu. Online data included both Chitwan and Kathmandu. Online data collection was conducted from 26 June 2021 to 24 October 2021, while paper-pencil data collection occurred from 27 September 2021 to 4 October 2021. Table 1 shows the details of the participants.

### 2.3. Measures

Two primary instruments were used for data collection, followed by a demographic information form.

The first item was the Mental Health Literacy Questionnaire (MHLq-young adult), a questionnaire which is a 29-item 5-point Likert scale to assess four dimensions of mental health literacy: knowledge of mental health problems (KMHP), erroneous beliefs/stereotypes (EBS), first aid and help-seeking behavior (FASHSB), and self-help strategies (SHS) [34]. The scale was found to have good Cronbach alpha (α = 0.84), suggesting that the scale can produce data with evidence of reliability. [32]. For the current study, Cronbach’s alpha was acceptable (α = 0.79), and high convergent validity was observed, with Pearson correlation (r) ranging from 0.60 to 0.77 between the subscales and the global scale.

The second tool was the Attitudes towards Mental Health Problems (ATMHP), which is a 35-item 4-point Likert scale to measure attitudes toward mental health problems across five dimensions: attitudes towards mental health problems (ATMHP), external shame (ES) (beliefs that others will look down on self if one has mental health problems/shame with family or community if any mental illness occurs), internal shame (IS) (shame related to negative self-evaluations/shame from within oneself), reflected shame 1 (RS1) (shame focused on the impact on the family if any mental illness occurs), and reflected shame 2 (RS2) (shame focused on the impact on oneself if any mental illness occurs) [29]. The scale demonstrated excellent Cronbach alpha (α = 0.94) for the reliability test [35]. This score is similar to the current study (α = 0.94). Moreover, high convergent validity was observed, with Pearson correlation (*r*) ranging from 66 to 86 between the subscales and the global scale.

### 2.4. Ethical Consideration

Ethical approval for the study was obtained from the Nepal Health Research Council (NHRC), ERB protocol registration no. 309/2021 (ref. no. 3543). Informed consent was secured from all participants, who were assured of their confidentiality and the voluntary nature of their participation. Though the data collection process involved minimal risks, including written consent and oral briefings, potential concerns about data security, coercion, and participant voluntariness remain. Participants were informed about their right to withdraw from the study at any time without consequence. The study adhered to the ethical principles outlined in the Declaration of Helsinki, ensuring the protection of participants’ rights throughout the research process [36].

### 2.5. Data Analysis

Data from Google Forms and paper-based surveys were merged after manually entering the paper-based responses into a matching Google Form. The combined dataset was processed using Google Sheets and MS Excel and then exported as CSV files for analysis in JASP. Data were analyzed descriptively to summarize participant characteristics and survey responses. Descriptive statistics were used to summarize demographics, while inferential tests, including correlations and group comparisons, were performed with a significance level of *p* < 0.05. Welch’s *t*-test was used to compare means between groups with unequal variances [37]. One-way Welch’s ANOVA with post hoc comparisons (Games Howell’s) assessed differences across multiple groups to address the unequal distribution of data [37,38]. Pearson’s correlation was used to examine the relationship between MHL and ATMHP [39]. Reliability was evaluated using Cronbach’s Alpha [40]. Additionally, we utilized Mendeley for citation management and ChatGPT for language editing and paraphrasing during the preparation of the manuscript.

## 3. Results

### 3.1. Demographic Components

The study comprised 385 participants aged 18 to 24 years, with the largest age group being 20 years old (20.52%). The gender comprised 61.3% females and 38.2% males, excluding other gender minorities. In the ethnic group, most participants were Brahmin/Kshetri (69.9%), followed by Janajaati (10.9%) and Newar (8.8%). In the religion category, 89.4% of participants identified as Hindu, while 10.4% belonged to other religions. In terms of academic qualifications, 84.4% of participants were pursuing a Bachelor’s degree, while 15.6% were pursuing an upper high school degree (class 11 and 12, age 18 and above). In fields of study, 34.8% were in Science, 29.1% in BS/BA/BM, 18.4% in other fields, and 16.4% in HM. Public institution students made up 46.5%, followed by government (31.7%) and private (21.8%) students, with 15.8% employed. Additionally, 59.5% did not know about PMHP, while 25.7% had some knowledge (Table 1).

### 3.2. Data Distribution

The dataset comprised 385 valid observations for two variables, the MHL and ATMHP. For MHL, the mean score was 119.51 (SD = 9.30), with a minimum score of 80 and a maximum score of 143. The data exhibited slight negative skewness (−0.71) and had a kurtosis of 1.75. Shapiro-Wilk test indicated a significant departure from normality (*p* < 0.001), and the test of equality of variance (Levene’s test) showed significant results, suggesting Welch’s correction of homogeneity in various comparison groups [37,41,42]. Regarding ATMHP, the mean score was 34.41 (SD = 19.79), with values ranging from 0 to 105. The data showed a slight positive skewness (0.58) and a kurtosis of −0.28. The Shapiro-Wilk test revealed significant deviation from normality (*p* < 0.001), and the test of equality of variance (Levene’s test) demonstrated significant results, suggesting Welch’s correction of homogeneity in various comparison groups [37,41,42].

### 3.3. Relationship Between MHL, ATMHP

The analysis showed no significant relationship between global MHL and global ATMHP, indicating no association between knowledge and attitudes. Minimal positive correlations were found between KMHP and ATMHP, KMHP and RS1, and KMHP and global ATMHP. Conversely, minimal negative correlations were observed between EBS and RS2, as well as FASHSB and IS. Strong positive correlations were noted between the global MHL and its factors, while strong to very strong correlations were observed between the global ATMHP and its factors, supporting the scales’ convergent validity (Table 2).

### 3.4. Connection Between MHL, ATMHP and Demographic Variables

No significant correlations with age were found (Table 2). The analysis revealed no significant gender differences in global MHL or its dimensions, except for the EBS factor, where females scored significantly higher than males (*p* = *0*.025, g = *0*.24). Likewise, no significant gender differences were observed in global ATMHP or its dimensions (Table 3).

No significant differences were found in global MHL or its dimensions among Brahmin/Kshetri, Janajati, Newar, and other groups. Similarly, no significant differences were observed in global ATMHP or its dimensions across these groups (Table 4).

No significant differences were found in global MHL or its dimensions between Hindu participants and those from other religions. Similarly, there were no significant differences in global ATMHP or its dimensions across these groups (Table 3).

No significant differences were found in global MHL or its dimensions between participants with a bachelor’s degree and those with a high school education. Similarly, no significant differences were observed in global ATMHP or its dimensions across these education levels (Table 3).

No significant differences were found in global MHL or three of its dimensions among participants from BS/BA/BM, HM, other, and science. However, a significant difference was observed in the SHS dimension (*p* = 0.049, *ω*^2^ = 0.12) with Games-Howell post hoc analysis indicating that HM participants scored significantly higher in SHS compared to others (*p* = 0.036, *d* = 0.46) (Table 4 and Table 5). Additionally, no significant differences were found in global ATMHP or its dimensions across these groups (Table 4).

No significant differences were found in global MHL among participants from government, private, and public colleges, except for the SHS factor (*p* = 0.007, *ω*^2^ = 0.02), where a significant difference was noted, suggesting a small effect size (Table 4). Games-Howell post hoc analysis showed that public college participants had significantly higher SHS levels compared to those from government (*p* = 0.036, *d* = −0.32) and private colleges (*p* = 0.02, *d* = −0.32), with a small effect size for both comparisons (Table 5). Additionally, no significant differences were observed in global ATMHP or its dimensions across these institutional groups (Table 4).

No significant differences were found in global MHL between employed and non-employed students, except for the SHS factor (*p* = 0.002, g = −0.46)**,** where a significant difference was observed. Additionally, no significant differences were found in global ATMHP between employed and non-employed students (Table 3).

Similarly, no significant differences were observed in global MHL or its dimensions between participants who know PMHP and those who do not. Likewise, no significant differences were found in global ATMHP or its dimensions between these groups (Table 3).

## 4. Discussion

Our study of 385 college students aimed to explore the connection between mental health literacy (MHL) and attitudes toward mental health problems (ATMHP) across diverse demographic groups. Contrary to expectations, we found that knowledge alone does not necessarily translate into more positive attitudes. This highlights a critical gap in mental health education efforts: simply increasing knowledge may not be enough to shift attitudes. These findings suggest that interventions targeting both knowledge and attitudes are essential to truly change perceptions of mental health issues among students.

We found no significant connection between MHL and ATMHP. This finding contradicts several previous studies that reported positive associations between MHL and mental health attitudes [6,7,8,9,22,23]. While Lopez et al. (2018) found that higher education correlated with greater depression knowledge and less stigma, they also noted an unexpected increase in stigma towards antidepressant use among more educated individuals [43]. This nuanced finding aligns with the observation that factors beyond knowledge, such as fear, insecurity, and unfavorable public image, contribute to negative attitudes [18]. The inconsistency between our findings and previous research is further complicated by studies showing that even healthcare professionals can exhibit negative attitudes towards individuals with mental illness [6,26,27,28]. This suggests that the link between MHL and ATMHP is not consistently positive across different populations and contexts. The intricate and often inconsistent interplay between mental health literacy, education, attitudes, and interpersonal relationships in the context of mental health underscores the need for holistic, culturally sensitive approaches that address not only knowledge gaps but also deeply rooted societal perceptions, fears, and relational dynamics to effectively combat stigma and improve mental health outcomes across diverse populations and social contexts. Our findings revealed comparable MHL between females and males across most dimensions, except for erroneous beliefs/stereotypes, where females showed significantly higher awareness. This partially aligns with previous research indicating higher MHL among females [10,11,16,44], though it diverges from a study reporting higher female knowledge in all dimensions except erroneous beliefs/stereotypes [34]. The literature presents a mixed landscape, with some studies reporting superior female performance in recognizing mental illnesses and recommending appropriate help [10,45], while others found poor mental health knowledge among women in certain contexts [46]. These inconsistencies in gender differences in mental health literacy emphasize the need for research that considers sociocultural factors and exposure to mental health information to guide the development of gender-specific mental health education programs.

The current study found that age had a non-significant connection with MHL. This finding is inconsistent with other studies [7,11,16,17]. We also found no connection between age and ATMHP. This result is in line with another study [12,47]. However, Lee et al. (2020) found older age is associated with lower mental health attitude levels in women [8].

Our study found that females exhibited a non-significantly higher level of negative attitudes towards mental health problems. This aligns with previous research that reported statistically non-significant gender differences in attitudes regarding mental illness research [12,13]. However, our findings contrast with studies that reported higher levels of negative attitudes among males [8,48,49]. Conversely, Neupane et al. (2016) found poorer attitudes among female caregivers [50]. These inconsistencies suggest that attitudes toward mental health problems are influenced by complex sociocultural factors. Notably, Gilbert et al. (2004) highlighted the significant impact of familial shame (izzat) on Asian women, indicating that fear of dishonoring others is closely tied to societal norms and cultural honor systems [51]. The varying results suggest that sociocultural factors influence attitudes toward mental health, emphasizing the need for research to guide culturally sensitive interventions and education programs.

We found no significant differences in MHL across groups, suggesting a universal consistency in views on certain disorders. This aligns with the finding of no significant distinctions between ethnic groups regarding major depressive disorder [52]; however, Marie et al. (2004) noted that ethnic backgrounds can influence mental health views. These findings indicate a need for further research to explore both universal and culturally specific perceptions to enhance mental health education strategies [52].

We also observed non-significant differences in ATMHP levels among ethnic groups. Similarly, Nepal et al. (2020) found no impact of ethnicity on attitude levels [48]. These findings suggest that ATMHP may be relatively consistent across ethnic groups in the context of Nepal.

We found no significant differences in MHL between Hindus and other participants, suggesting religious affiliation may not affect overall knowledge. However, some religious communities may interpret mental health issues through a spiritual lens, influencing literacy and treatment approaches [14,53]. These findings highlight the need for further research to better understand how diverse religious perspectives MHL is.

Similarly, we found no association between religious beliefs and ATMHP in Nepal, which is consistent with a study [48]. Some religious contexts may view mental health issues as a test of faith, leading to a preference for spiritual over medical treatment [54]. While Wesselmann et al. (2010) found a link between religiosity and lower mental health stigma [53], Cinnirella (1999) reported mixed results [15]. Further research is needed to clarify these relationships.

We observed no significant differences in MHL were observed across academic levels in the current study. However, the students with low education exhibited low levels of MHL [19,43], which is consistent with findings that mental health education and higher education levels positively correlate with MHL [16,17] and academic advisors in teaching roles with greater mental health literacy than their non-teaching counterparts [21]. These mixed results underscore the need for large-scale studies to clarify the relationship between academic levels and MHL.

We found no significant differences in ATMHP across academic levels, consistent with the findings of Salve (2013) [13]. However, studies also indicate that higher education is associated with less stigmatizing attitudes [7,48], though stigma persists even among educated individuals [12]. Higher education among caregivers and health workers is linked to more positive attitudes [26,28,43,50]. Risal et al. (2013) found that medical students and interns generally held positive or neutral attitudes toward mental illness and psychiatry [55]. These mixed findings suggest the need for further research to clarify the relationship between education and attitudes toward mental illness.

The difference was non-significant in MHL across study fields; however, we observed significantly higher levels of self-help strategies among students from hotel management compared to students who categorized themselves as part of the other group. The distinction was non-significant among the rest of the comparison groups. We observed no significant differences in overall MHL across study fields. However, Al-Atram (2018) found a statistically significant relationship between specialty and knowledge [56]. These inconsistencies highlight the need for further research to clarify the role of knowledge in mental health literacy.

We observed no significant differences in ATMHP across study fields. Studies indicate that healthcare professionals hold varying attitudes toward mental health issues. For instance, nurses with higher education levels often exhibited authoritarian views [57,58], while medical students generally viewed mental illness similarly to other medical conditions and had positive attitudes toward psychiatry [49,55]. However, some medical students held negative attitudes, with many believing that individuals with mental illness were more likely to harm others [59]. These findings suggest that, while no significant differences were observed among study fields, there are diverse understandings and attitudes within healthcare professions.

Participants from public colleges had significantly higher awareness of self-help strategies (SHS) compared to those from government and private colleges. However, no significant differences were found in mental health literacy across types of institutions for other dimensions. The reason for the higher awareness in public colleges remains unclear and requires further investigation.

The lack of significant differences in ATMHP across institutional groups may indicate a convergence of attitudes shaped by overarching cultural and societal influences. Research has shown that attitudes toward mental health are often influenced by collective societal norms rather than isolated institutional practices [51]. This phenomenon indicates that irrespective of the institutional setting—whether in healthcare, education, or community-based environments—individuals may develop similar stigmas and perceptions related to mental health. Non-employed students showed higher knowledge of self-help strategies. However, employment status did not significantly affect overall mental health literacy, consistent with findings that other factors may be more influential [60,61]. Working students in Nepal with flexible attendance may have less access to resources compared to non-employed students who may attend and engage in college activities. Engagement with mental health resources is influenced by their availability and perceived usefulness [62]. The increased knowledge among non-employed students may result from their greater ability to allocate time to self-help activities, which employed students might lack due to work commitments.

Similarly, employment status did not significantly affect students’ overall attitude levels. This result suggests that attitude may be more influenced by individual beliefs and societal factors than by immediate contextual factors like employment status [2,63].

We found no significant difference in MHL levels between students who know PMHP and those who do not. Studies have shown that knowing someone with a mental health issue or having a personal or family history of mental disorders is associated with higher MHL levels [7,34,44,64]. These findings suggest that while personal connections and history may enhance MHL, they do not lead to significant differences across broader populations. Further research is needed to explore factors contributing to MHL development.

We found no significant difference in attitudes between students who know PMHP and those who do not. Perceptions of mental illness vary, with many holding stigmatizing attitudes towards PMHP regarding treatment, work, marriage, and recovery [44]. General populations and specialists often have negative attitudes toward psychiatric patients, while family practitioners tend to be more positive [6,56], which is consistent with familiarity with close individuals with mental illness is linked to less stigmatizing attitudes, whereas knowing non-close individuals is associated with less favorable behaviors [7]. These findings highlight the complexity of attitudes toward mental illness, with personal connections and professional roles playing key roles. Further research is needed to explore these dynamics and reduce negative attitudes.

### 4.1. Implication

Our study highlights the critical need for community initiatives and interventions in Nepal that go beyond merely enhancing mental health literacy (MHL) to effectively address attitudes toward mental health problems (ATMHP). The findings emphasize that increasing knowledge alone is insufficient to reduce stigma; instead, culturally sensitive approaches are essential to challenge deeply rooted societal perceptions and misconceptions. Interventions should integrate strategies that address social and cultural influences, including fear, insecurity, and norms while fostering open discussions. Furthermore, the development of gender-specific mental health education programs may help address mixed findings on gender differences in MHL and ATMHP. These insights can inform educators, policymakers, and mental health professionals in designing inclusive programs that encourage help-seeking behavior, reduce stigma, and ultimately improve mental health outcomes across diverse communities.

### 4.2. Limitations

This study has several limitations that may influence its findings and generalizability. The focus on Nepalese upper high schools (class 11 and 12) and bachelor’s students (aged 18–24) from urban areas restricts applicability to other age groups, illiterate populations, and students from remote regions, such as Karnali. The cultural adaptation of instruments and lack of invariance testing further limit conclusions about their applicability across diverse demographic groups, and the categorization of gender as male and female excludes representation of diverse gender identities. The timing of the study during the 2021 COVID-19 pandemic may have introduced unique stressors affecting participants’ mental health literacy and attitudes. Additionally, reliance on self-reported data introduces bias, and the cross-sectional design limits causal inferences. Future studies should address these gaps to provide a more comprehensive understanding of mental health literacy and attitudes across diverse populations.

### 4.3. Future Direction

Based on the current findings, future research should address several key areas. Given that MHL does not appear related to ATMHP, further investigation is needed to understand this discrepancy. Specifically, research should explore why females exhibit lower levels of erroneous beliefs and stereotypes, why students from hotel management and public college students employ more self-help strategies, and how working status affects mental health attitudes. Additionally, future research should include more inclusive categories to better capture the impact of gender diversity on mental health literacy and attitudes toward mental health problems. A longitudinal study could provide deeper insights into these variables and their evolving impact on ATMHP over time.

## Figures and Tables

**Table 1 behavsci-14-01189-t001:** Demographic Characteristics.

Demographics	Frequency	Percent	Demographics	Frequency	Percent
Gender (Aligned with Biological Sex)			Field of Study		
Female	236	61.30	BS/BA/BM	112	29.09
Male	147	38.18	BM	63	16.36
Missing	2	0.52	Others	71	18.44
Total	385	100	Science	134	34.81
Ethnicity			Missing	5	1.30
Brahmin/Kshetri	269	69.87	Total	385	100
Janajaati	42	10.91	Type of Institution		
Newar	34	8.83	Government	122	31.69
Others	36	9.35	Private	84	21.82
Missing	4	1.04	Public	179	46.49
Total	385	100	Missing	0	0
Religion			Total	385	100
Hindu	344	89.35	Work Status		
Others	40	10.39	Employed Students	61	15.84
Missing	1	0.26	Student	323	83.90
Total	385	100	Missing	1	0.26
Academic Qualification			Total	385	100
Bachelor’s Degree	325	84.42	Participants who do not know PMHP		
High School Degree	60	15.58	Participants who do not know PMHP	229	59.48
Missing	0	0	Participants who know PMHP	99	25.71
Total	385	100	Missing	57	14.81
			Total	385	100

Note: BS = business studies, BA = business administration, BM = business management, HM = hotel management, and PMHP = people with mental health problems.

**Table 2 behavsci-14-01189-t002:** Pearson Correlation.

Variables		1	2	3	4	5	6	7	8	9	10	11
1. Age	Pearson’s r	—										
	*p*-value	—										
2. KMHP Factor	Pearson’s r	0.02	—									
	*p*-value	0.761	—									
3. EBS Factor	Pearson’s r	0.01	0.21 ***	—								
	*p*-value	0.85	<0.001	—								
4. FASHSB Factor	Pearson’s r	0.09	0.41 ***	0.32 ***	—							
	*p*-value	0.084	<0.001	<0.001	—							
5. SHS Factor	Pearson’s r	−0.14 **	0.37 ***	0.21 ***	0.38 ***	—						
	*p*-value	0.006	<0.001	<0.001	<0.001	—						
6. Global MHL	Pearson’s r	0.01	0.77 ***	0.67 ***	0.73 ***	0.60 ***	—					
	*p*-value	0.857	<0.001	<0.001	<0.001	<0.001	—					
7. ATMHP Factor	Pearson’s r	0.07	0.14 **	−0.07	−0.06	−0.05	0.01	—				
	*p*-value	0.166	0.006	0.184	0.211	0.34	0.92	—				
8. ES Factor	Pearson’s r	−0.05	0.09	−0.05	−0.09	0.01	−0.01	0.55 ***	—			
	*p*-value	0.302	0.076	0.349	0.071	0.922	0.897	<0.001	—			
9. IS Factor	Pearson’s r	0.04	0.06	−0.02	−0.12 *	0.04	−0.01	0.29 ***	0.52 ***	—		
	*p*-value	0.438	0.26	0.721	0.025	0.452	0.854	<0.001	<0.001	—		
10. RS1 Factor	Pearson’s r	−0.07	0.10 *	−0.02	−0.03	0.05	0.04	0.41 ***	0.62 ***	0.57 ***	—	
	*p*-value	0.145	0.049	0.732	0.526	0.315	0.433	<0.001	<0.001	<0.001	—	
11. RS2 Factor	Pearson’s r	−0.06	0	−0.11 *	−0.07	0.05	−0.06	0.20 ***	0.38 ***	0.49 ***	0.53 ***	—
	*p*-value	0.248	0.967	0.038	0.145	0.333	0.27	<0.001	<0.001	<0.001	<0.001	—
12. Global ATMHP	Pearson’s r	−0.03	0.11 *	−0.07	−0.1	0.02	−0.01	0.68 ***	0.86 ***	0.72 ***	0.83 ***	0.66 ***
	*p*-value	0.611	0.034	0.182	0.056	0.679	0.921	<0.001	<0.001	<0.001	<0.001	<0.001

* *p* < 0.05, ** *p* < 0.01, *** *p* < 0.001.

**Table 3 behavsci-14-01189-t003:** Welch’s *t*-test Results.

Categories	Group	N	Mean	SD	*t*	df	*p*
Gender							
KMHP Factor	Female	236	43.27	3.92	0.54	262.03	0.588
	Male	147	43.01	4.85			
EBS Factor	Female	236	33.81	3.63	2.26	275.86	0.025 * (g = 0.24)
	Male	147	32.86	4.21			
FASHSB Factor	Female	236	25.43	2.67	−0.68	253.64	0.5
	Male	147	25.65	3.45			
SHS Factor	Female	236	17.39	1.93	−0.06	280.23	0.949
	Male	147	17.41	2.2			
Global MHL	Female	236	119.9	8.34	0.94	256.17	0.35
	Male	147	118.93	10.63			
ATMHP Factor	Female	236	8.95	5.23	−0.39	296.21	0.695
	Male	147	9.18	5.54			
ES Factor	Female	236	8.42	6.82	0.32	289.46	0.753
	Male	147	8.18	7.44			
IS Factor	Female	236	3.52	3.54	−0.86	284.43	0.391
	Male	147	3.86	3.95			
RS1 Factor	Female	236	7.72	5.23	−0.25	297.30	0.801
	Male	147	7.86	5.51			
RS2 Factor	Female	236	5.79	4.51	1.33	297.96	0.183
	Male	147	5.14	4.74			
Global ATMHP	Female	236	34.4	18.82	0.08	281.62	0.933
	Male	147	34.22	21.26	0.08	281.6	0.93
Religion							
KMHP Factor	Hindu	344	43.22	4.33	0.34	49.00	0.733
	Others	40	42.98	4.23			
EBS Factor	Hindu	344	33.41	3.91	−0.42	49.20	0.674
	Others	40	33.68	3.78			
FASHSB Factor	Hindu	344	25.58	3.01	1.70	50.11	0.095
	Others	40	24.78	2.80			
SHS Factor	Hindu	344	17.40	2.04	0.15	48.50	0.882
	Others	40	17.35	2.05			
Global MHL	Hindu	344	119.60	9.40	0.57	50.62	0.57
	Others	40	118.78	8.55			
ATMHP Factor	Hindu	344	8.97	5.21	−0.85	45.34	0.403
	Others	40	9.85	6.34			
ES Factor	Hindu	344	8.34	7.08	−0.16	49.43	0.874
	Others	40	8.53	6.78			
IS Factor	Hindu	344	3.62	3.71	−0.75	47.81	0.459
	Others	40	4.10	3.85			
RS1 Factor	Hindu	344	7.82	5.35	0.37	49.14	0.712
	Others	40	7.50	5.19			
RS2 Factor	Hindu	344	5.60	4.65	0.44	50.16	0.661
	Others	40	5.28	4.31			
Global ATMHP	Hindu	344	34.36	19.72	−0.26	47.61	0.796
	Others	40	35.25	20.70			
Academic Level							
KMHP Factor	Bachelor’s Degree	325	43.19	4.4	0.07	90.62	0.945
	High School	60	43.15	3.81			
EBS Factor	Bachelor’s Degree	325	33.58	3.91	1.74	85.45	0.085
	High School	60	32.67	3.68			
FASHSB Factor	Bachelor’s Degree	325	25.55	3.02	0.82	85	0.413
	High School	60	25.22	2.87			
SHS Factor	Bachelor’s Degree	325	17.33	2.07	−1.52	88.86	0.132
	High School	60	17.73	1.84			
Global MHL	Bachelor’s Degree	325	119.65	9.34	0.69	83.58	0.496
	High School	60	118.77	9.11			
ATMHP Factor	Bachelor’s Degree	325	9.23	5.31	1.48	81.39	0.142
	High School	60	8.1	5.41			
ES Factor	Bachelor’s Degree	325	8.47	6.92	0.72	77.71	0.474
	High School	60	7.7	7.68			
IS Factor	Bachelor’s Degree	325	3.81	3.77	1.87	88.98	0.065
	High School	60	2.92	3.35			
RS1 Factor	Bachelor’s Degree	325	7.91	5.32	1.03	82.21	0.306
	High School	60	7.13	5.33			
RS2 Factor	Bachelor’s Degree	325	5.6	4.59	0.4	80.58	0.693
	High School	60	5.33	4.76			
Global ATMHP	Bachelor’s Degree	325	35	19.57	1.32	79.42	0.191
	High School	60	31.18	20.82			
Work Status							
KMHP Factor	Employed Students	61	42.85	4.50	−0.65	81.78	0.518
	Not employed students	323	43.26	4.28			
EBS Factor	Employed Students	61	32.85	3.92	−1.26	83.65	0.211
	Not employed students	323	33.54	3.88			
FASHSB Factor	Employed Students	61	25.07	3.61	−1.04	74.88	0.3
	Not employed students	323	25.58	2.86			
SHS Factor	Employed Students	61	16.61	2.10	−3.23	81.80	0.002 ** (g = −0.46)
	Not employed students	323	17.55	2.00			
Global MHL	Employed Students	61	117.38	10.19	−1.82	79.08	0.073
	Not employed students	323	119.92	9.10			
	Student	323	119.92	9.10			
ATMHP Factor	Employed Students	61	8.80	6.09	−0.37	77.28	0.709
	Not employed students	323	9.12	5.19			
ES Factor	Employed Students	61	7.69	7.86	−0.71	78.38	0.479
	Not employed students	323	8.46	6.89			
IS Factor	Employed Students	61	3.77	4.17	0.21	78.22	0.833
	Not employed students	323	3.65	3.64			
RS1 Factor	Employed Students	61	7.03	5.16	−1.26	86.08	0.211
	Not employed students	323	7.95	5.35			
RS2 Factor	Employed Students	61	5.34	5.02	−0.39	79.55	0.698
	Not employed students	323	5.61	4.53			
Global ATMHP	Employed Students	61	32.64	22.36	−0.70	77.78	0.486
	Not employed students	323	34.78	19.30			
Participants who know PMHP and Participants who do not know PMHP							
KMHP Factor	Participants who know PMHP	99	43.59	3.88	−1.03	222.48	0.305
	Participants who do not know PMHP	229	43.07	4.68			
EBS Factor	Participants who know PMHP	99	33.84	3.71	−1.24	203.41	0.218
	Participants who do not know PMHP	229	33.27	4.08			
FASHSB Factor	Participants who know PMHP	99	25.57	2.94	−0.43	195.20	0.666
	Participants who do not know PMHP	229	25.41	3.1			
SHS Factor	Participants who know PMHP	99	17.36	1.79	0.35	227.58	0.724
	Participants who do not know PMHP	229	17.45	2.21			
Global MHL	Participants who know PMHP	99	120.35	8.17	−1.09	228.91	0.278
	Participants who do not know PMHP	229	119.2	10.16			
ATMHP Factor	Participants who know PMHP	99	9.4	5.42	−0.6	182.29	0.546
	Participants who do not know PMHP	229	9.01	5.29			
ES Factor	Participants who know PMHP	99	9.17	7.46	−1.33	173.55	0.187
	Participants who do not know PMHP	229	8.01	6.9			
IS Factor	Participants who know PMHP	99	3.35	3.51	1.14	204.51	0.255
	Participants who do not know PMHP	229	3.85	3.88			
RS1 Factor	Participants who know PMHP	99	7.97	5.38	−0.51	185.57	0.612
	Participants who do not know PMHP	229	7.64	5.36			
RS2 Factor	Participants who know PMHP	99	4.9	4.43	1.94	195.26	0.053
	Participants who do not know PMHP	229	5.95	4.67			
Global ATMHP	Participants who know PMHP	99	34.8	20.03	−0.14	184.52	0.891
	Participants who do not know PMHP	229	34.47	19.84			

* *p* < 0.05, ** *p* < 0.01. Note: ‘g’ refers to ‘Hedges’ g’, and PMHP refers to people with mental health problems.

**Table 4 behavsci-14-01189-t004:** One-way ANOVA with Welch’s Homogeneity Correction.

Variables	N	Mean	SD	SE	df	F	*p*
Ethnicity							
KMHP Factor							
Brahmin/Kshetri	269	43.07	4.01	0.25	3, 69.57	0.34	0.796
Janajaati	42	43.64	4.29	0.66			
Newar	34	42.94	5.89	1.01			
Others	36	43.61	4.81	0.80			
EBS Factor							
Brahmin/Kshetri	269	33.54	3.95	0.24	3, 77.12	1.74	0.166
Janajaati	42	32.76	4.50	0.70			
Newar	34	32.77	2.82	0.48			
Others	36	34.28	3.38	0.56			
FASHSB Factor							
Brahmin/Kshetri	269	25.54	3.11	0.19	3, 76.14	0.28	0.839
Janajaati	42	25.24	2.75	0.42			
Newar	34	25.29	2.75	0.47			
Others	36	25.72	2.74	0.46			
SHS Factor							
Brahmin/Kshetri	269	17.31	1.94	0.12	3, 71.03	0.91	0.443
Janajaati	42	17.69	1.98	0.31			
Newar	34	17.21	2.87	0.49			
Others	36	17.75	1.95	0.33			
Global MHL							
Brahmin/Kshetri	269	119.47	9.23	0.56	3, 73.91	0.91	0.439
Janajaati	42	119.33	10.03	1.55			
Newar	34	118.21	11.25	1.93			
Others	36	121.36	7.12	1.19			
ATMHP Factor							
Brahmin/Kshetri	269	9.04	5.43	0.33	3, 74.38	1.27	0.292
Janajaati	42	10.36	5.02	0.78			
Newar	34	8.38	5.09	0.87			
Others	36	8.44	5.40	0.90			
ES Factor							
Brahmin/Kshetri	269	8.09	7.15	0.44	3, 74.29	0.59	0.623
Janajaati	42	9.19	6.24	0.96			
Newar	34	9.38	7.39	1.27			
Others	36	8.50	6.99	1.17			
IS Factor							
Brahmin/Kshetri	269	3.51	3.79	0.23	3, 75.69	1.44	0.239
Janajaati	42	4.74	3.93	0.61			
Newar	34	3.97	3.21	0.55			
Others	36	3.28	3.27	0.54			
RS1 Factor							
Brahmin/Kshetri	269	7.78	5.44	0.33	3, 74.80	0.5	0.687
Janajaati	42	8.57	4.67	0.72			
Newar	34	7.41	5.05	0.87			
Others	36	7.42	5.59	0.93			
RS2 Factor							
Brahmin/Kshetri	269	5.46	4.72	0.29	3, 74.87	0.49	0.692
Janajaati	42	6.14	4.00	0.62			
Newar	34	6.03	4.44	0.76			
Others	36	5.28	4.81	0.80			
Global ATMHP							
Brahmin/Kshetri	269	33.88	20.14	1.23	3, 74.71	0.96	0.418
Janajaati	42	39.00	19.03	2.94			
Newar	34	35.18	19.48	3.34			
Others	36	32.92	18.27	3.04			
Field of Study (in bachelor’s degree							
KMHP Factor							
BS/BA/BM	112	42.97	3.63	0.34	3, 171.99	0.23	0.873
HM	63	43.46	4.93	0.62			
Others	71	42.99	4.39	0.52			
Science	134	43.28	4.55	0.39			
EBS Factor							
BS/BA/BM	112	33.57	3.52	0.33	3, 172.57	1.34	0.264
HM	63	32.29	5.34	0.67			
Others	71	33.59	3.14	0.37			
Science	134	33.78	3.73	0.32			
FASHSB Factor							
BS/BA/BM	112	25.48	2.60	0.25	3, 171.46	0.01	0.999
HM	63	25.52	3.57	0.45			
Others	71	25.55	3.01	0.36			
Science	134	25.49	3.07	0.27			
SHS Factor							
BS/BA/BM	112	17.27	1.97	0.19	3, 178.74	2.66	0.049 *
HM	63	17.92	2.03	0.26			(*ω*^2^ = 0.12)
Others	71	16.99	1.92	0.23			
Science	134	17.46	2.15	0.19			
Global MHL							
BS/BA/BM	112	119.30	7.43	0.70	3, 172.04	0.19	0.902
HM	63	119.19	11.24	1.42			
Others	71	119.11	8.65	1.03			
Science	134	120.00	10.21	0.88			
ATMHP Factor							
BS/BA/BM	112	9.47	5.49	0.52	3, 177.23	1.15	0.331
HM	63	8.13	4.87	0.61			
Others	71	8.62	5.73	0.68			
Science	134	9.25	5.20	0.45			
ES Factor							
BS/BA/BM	112	8.37	7.35	0.69	3, 174.2	0.06	0.979
HM	63	8.08	6.90	0.87			
Others	71	8.20	7.66	0.91			
Science	134	8.50	6.60	0.57			
IS Factor							
BS/BA/BM	112	4.33	3.75	0.36	3, 173.91	1.79	0.151
HM	63	3.43	3.84	0.48			
Others	71	3.48	3.86	0.46			
Science	134	3.29	3.53	0.31			
RS1 Factor							
BS/BA/BM	112	8.39	5.12	0.48	3, 176.28	1.08	0.361
HM	63	7.10	5.32	0.67			
Others	71	7.28	5.41	0.64			
Science	134	7.81	5.40	0.47			
RS2 Factor							
BS/BA/BM	112	6.22	4.25	0.40	3, 173.56	1.47	0.225
HM	63	5.25	4.39	0.55			
Others	71	5.86	5.42	0.64			
Science	134	5.13	4.51	0.39			
Global ATMHP							
BS/BA/BM	112	36.79	19.58	1.85	3, 172.48	0.91	0.436
HM	63	31.98	20.33	2.56			
Others	71	33.44	21.92	2.60			
Science	134	33.97	18.60	1.61			
Types of Institute							
KMHP Factor							
Government	122	43.14	4.89	0.44	2, 217.09	0.91	0.403
Private	84	43.66	3.55	0.39			
Public	179	42.99	4.23	0.32			
EBS Factor							
Government	122	33.21	3.72	0.34	2, 235.37	1.11	0.331
Private	84	33.86	2.82	0.31			
Public	179	33.39	4.39	0.33			
FASHSB Factor							
Government	122	25.05	3.47	0.31	2, 197.12	1.82	0.164
Private	84	25.86	2.93	0.32			
Public	179	25.64	2.63	0.20			
SHS Factor							
Government	122	17.10	2.39	0.22	2, 206.11	5.16	0.007 **
Private	84	17.10	1.77	0.19			(*ω*^2^ = 0.02)
Public	179	17.74	1.84	0.14			
Global MHL							
Government	122	118.49	11.36	1.03	2, 218.78	1.2	0.304
Private	84	120.46	6.87	0.75			
Public	179	119.75	8.70	0.65			
ATMHP Factor							
Government	122	9.07	5.62	0.51	2, 199.44	0.28	0.753
Private	84	9.41	5.65	0.62			
Public	179	8.87	5.00	0.37			
ES Factor							
Government	122	9.06	7.54	0.68	2, 184.04	2.68	0.071
Private	84	9.17	8.45	0.92			
Public	179	7.48	5.80	0.43			
IS Factor							
Government	122	3.30	3.59	0.33	2, 193.03	1.79	0.169
Private	84	4.41	4.44	0.48			
Public	179	3.58	3.39	0.25			
RS1 Factor							
Government	122	8.25	5.78	0.52	2, 193.07	0.82	0.444
Private	84	7.81	5.84	0.64			
Public	179	7.45	4.72	0.35			
RS2 Factor							
Government	122	5.50	4.92	0.45	2, 199.86	0.7	0.498
Private	84	5.10	4.79	0.52			
Public	179	5.81	4.30	0.32			
Global ATMHP							
Government	122	35.19	20.53	1.86	2, 187.88	0.66	0.517
Private	84	35.88	23.75	2.59			
Public	179	33.18	17.09	1.28			

* *p* < 0.05, ** *p* < 0.01. Note: ‘*ω*^2^’ refers to ‘omega squire’ for effect size. BS = business studies, BA = business administration, BM = business management, HM = hotel management.

**Table 5 behavsci-14-01189-t005:** Games Howell Post Hoc Comparisons (Tukey’s HSD) for Types of Education SHS factor in MHL.

		95% CI for Mean Difference					
Comparison	Mean Difference	Lower	Upper	SE	*t*	df	*p*-Value
SHS factor in Field of Study							
BS/BA/BM—HM	−0.65	−1.48	0.17	0.32	−2.07	125.51	0.17
BS/BA/BM—Others	0.28	−0.48	1.05	0.29	0.96	152.06	0.772
BS/BA/BM—Science	−0.20	−0.88	0.49	0.26	−0.74	241.98	0.88
HM—Others	0.94	0.05	1.83	0.34	2.73	128.03	0.036 * (*d* = 0.46)
HM—Science	0.46	−0.36	1.28	0.32	1.45	128.30	0.471
Others—Science	−0.48	−1.24	0.29	0.29	−1.62	157.69	0.368
SHS factor in Types of Institutes							
Government—Private	0.00	−0.68	0.69	0.29	0.01	202.84	1
Government—Public	−0.64	−1.24	−0.03	0.26	−2.49	214.23	0.036 * (*d* = −0.32)
Private—Public	−0.64	−1.20	−0.08	0.24	−2.71	167.73	0.02 * (*d* = −0.32)

* *p* < 0.05. Note: ‘d’ refers to ‘Cohen’s d’ for effect size. BS = business studies, BA = business administration, BM = business management, HM = hotel management.

## Data Availability

Certain data is available upon reasonable request from the corresponding author.

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
