# Peer review of "Mental Health Literacy and Attitudes Towards Mental Health Problems Among College Students, Nepal"

_behavsci, 2024, doi:10.3390/bs14121189_

Round 1
Reviewer 1 Report
Comments and Suggestions for Authors
Dear Authors,
Thank you for your work on this important topic. The study addresses a critical area of research by exploring mental health literacy (MHL) and attitudes toward mental health problems (ATMHP) in the Nepali context, which is an underrepresented population in current literature. Your contribution has the potential to provide meaningful insights and drive future work in this area. Below, I offer some suggestions to further strengthen your manuscript:
Theoretical Foundation:
The introduction provides a solid starting point by defining key concepts. However, it would benefit from a deeper exploration of how MHL and ATMHP interact within a theoretical framework. Additionally, including references to studies conducted in populations culturally or contextually similar to Nepal would enhance the contextualization of your work.
Instrument Validation:
One key consideration is whether the instruments used in your study, such as the Mental Health Literacy Questionnaire and the Attitudes Towards Mental Health Problems scale, have been validated specifically for the Nepali population or similar cultural settings. Clarifying this would strengthen the reliability and applicability of your findings.
Measurement Invariance:
Since your manuscript compares groups based on demographic variables, it is crucial to test for measurement invariance. This step ensures that the instruments measure the same constructs across groups and that any observed differences are genuine rather than artifacts of the measurement process. Addressing this aspect would add rigor and credibility to your comparisons.
Discussion Section:
The discussion is clear and well-written, but it could be enhanced by expanding on the practical implications of your findings, particularly for mental health education, policy-making, or interventions in Nepal. Additionally, acknowledging potential limitations, such as the cultural adaptation of instruments or the lack of invariance testing, would provide a balanced perspective.
Language and Clarity:
While the language used is clear and conveys the research effectively, minor revisions could improve readability and flow, ensuring that the presentation matches the quality of the research.
Round 1
Comment 1: The introduction provides a solid starting point by defining key concepts. However, it would benefit from a deeper exploration of how MHL and ATMHP interact within a theoretical framework. Additionally, including references to studies conducted in populations culturally or contextually similar to Nepal would enhance the contextualization of your work.
Response 1: Agree. We have now added the framework in the last paragraph of the introduction in the manuscript. However, there are very limited studies in the area of study in similar context of Nepal.
Comment 2: One key consideration is whether the instruments used in your study, such as the Mental Health Literacy Questionnaire and the Attitudes towards Mental Health Problems scale, have been validated specifically for the Nepali population or similar cultural settings. Clarifying this would strengthen the reliability and applicability of your findings.
Response 2: Agreed. We employed the tools which are well developed and validated in different cultural context. To address the reliability and validity concern of the tools, we translated the tool in Nepali according to tool translation protocols and calculated their reliability scores and convergent validity.
Comments 3: Since your manuscript compares groups based on demographic variables, it is crucial to test for measurement invariance. This step ensures that the instruments measure the same constructs across groups and that any observed differences are genuine rather than artifacts of the measurement process. Addressing this aspect would add rigor and credibility to your comparisons.
Response 3: Your concern is genuine. However, we were unable to conduct measurement invariance analyses due to limited data points and unequal group sizes. Our primary objective was to explore the relationship between MHL and ATMHP, and as such, we focused on ensuring the validity of our findings through alternative approaches. Specifically, we conducted homogeneity of variance tests and employed Welch’s t-test where necessary to address potential variability. Additionally, we assessed the reliability of the instrument using [e.g., Cronbach’s alpha] and examined factor correlations to establish convergent validity. We will certainly address this concern in the limitations section of the discussion in the manuscript.
Comment 4: The discussion is clear and well-written, but it could be enhanced by expanding on the practical implications of your findings, particularly for mental health education, policy-making, or interventions in Nepal. Additionally, acknowledging potential limitations, such as the cultural adaptation of instruments or the lack of invariance testing, would provide a balanced perspective.
Response 4: Agreed. We have improved the section which have been highlighted in green letters.
Comment 5: While the language used is clear and conveys the research effectively, minor revisions could improve readability and flow, ensuring that the presentation matches the quality of the research.
Response 5: Agreed. We have made efforts in a few areas. The areas have been green highlighted.
Please note that all the changes have been highlighted in green letters so that you can see them without much effort.
Reviewer 2 Report
Comments and Suggestions for Authors
Thank you for the opportunity to review this paper. Please see attached for comments and feedback for the piece.

Round 2
Introduction
Comment 1: Page 2 line reliable data, change to “data with evidence of reliability”
Response 1: Agreed. Corrected
Materials and Methods
Comment 2: The first and second paragraph seem to have some overlap with the wording. Please Condense into one seamless paragraph to reduce redundancy.
Response 2: Agreed. Corrected
Comment 3: Relationship describes a connection between two people, relation describes a connection between to variables (page 3 line 97). This was seen elsewhere in the manuscript, please
Response 3: Agreed. Correct throughout.
Comment 4: Facebook, (comma instead of a period) and comma needed after Instagram, (page 3 line
113)
Response 4: Agreed. Corrected
Comment 5: “were utilized to access online” (page 3 line 115), something is missing here.
Response 5: Agreed. Added
Comment 6: “The primary investigators collected the data from both online and paper-based methods.” (page 3 line 116) is redundant, please remove
Response 6: Agreed. Removed
Comment 7: Data is plural (page 3 line 118) so it would be “were”
Throughout the manuscript, but especially in the participants and procedure section, verb tense switched frequently and was harder to read, please revise
Response 7: Agreed. Corrected
Comment 8: The authors note “gender” as a header in Table 1, but within the subgroups, refer to terms that are not gender but rather categories of sex (female/male). Additionally, were data collected for individuals who identified as gender non-conforming?
If not, please explain why.
Response 8: Agreed. Specified with relevant information: the issue has been addressed with necessary information in different sections such as methods (i.e. participants and procedure), results and limitation sections.
Measures
Comment 9: Page 4 line 132, scales cannot be “reliable” but can produce data with evidence of reliability, please correct.
Response 9: Agreed. Corrected
Results
Comment 10: Awkward phrases “the gender included” (page 5 line 175):
Response 10: Agreed. Corrected
Comment 11: Awkward phrases “the majority in religion” (page 5 line 177)
Response 11: Agreed. Corrected
Comment 12: Awkward phrasing “holding a Bachelor’s degree” (page 5 line 178), perhaps “held a
Bachelor’s degree”
Response 12: Agreed. Corrected
Comment 13: The detail is given in appendix (Table 1) (page 5 line 183) – I am not sure what this means, please reword
Response 13: Agreed. Removed
Comment 14: Table 2 spelling error in “Correlation”
Response 14: Agreed. Corrected
Comment 15: The Tables are very comprehensive (Tables 3 and 4), but are hard to follow in the text because they are so large. Should these tables be put in appendices? Or perhaps converted to one table to fit on a landscape oriented page?
Response 15: Agreed. Replaced in appendix
Comment 16: For the significant findings, please also reference the effect sizes in the text to discuss practical significance.
Response 16: Agreed. Added
Discussion
Comment 17: Who is “they” (page 15 line 260; page 16 line 303)?
Response 17: Agreed. Addressed with author’s name.
Comment 18: In the discussion there is some confusion with sex and gender, while the text mentions the word gender, terms of sex “male and female” are used. Are the researchers examining sex or gender as their overall construct?
Response 18: Agreed. The issue has been addressed with necessary information in different sections such as methods (i.e. participants and procedure), results and limitation sections.
Comment 19: “However, other study found different scenario” (page 16 line 286) – the sentence missing something
Response 19: Agreed. Addressed the issue.
Comment 20: The word “lower” is a comparison word, so there needs to be a group that is compared (page 16 line 321)
Response 20: Agreed. Corrected
Comment 21: “We found no significant differences in ATMHP across academic levels, consistent with findings from a study” (page 16 line 327) – this sentence is missing something at the end to describe the findings from “a study”
Response 21: Agreed. Corrected
Comment 22: Oftentimes in the discussion “a study” is used instead of referencing the authors directly
Response 22: Agreed. Corrected
Comment 23: Please eliminate contractions throughout, for example, don’t on page 17 line 378
Response 23: Agreed. Corrected
Comment 24: Overall, the discussion section was very interesting but easy to get lost in given the sheer number of variables and non-significant findings. Perhaps the authors should consider a more organized manner to present the results in, such as in a table or figure, or with headers. Specifically, headers for the limitations and future directions would be helpful to add.
Response 24: Agreed. Addressed the issue
Round 2
Reviewer 1 Report
Comments and Suggestions for Authors
I sincerely appreciate the effort you have devoted to respond to each of the comments made. The advances in the manuscript are remarkable and enrich your work.
The inclusion of the theoretical framework in the introduction provides a solid basis for understanding the interaction between MHL and ATMHP. I also appreciate the clarity with which they explained the validation of the instruments and how they addressed reliability and validity in the Nepalese context.
I especially emphasize the honesty in acknowledging the limitations in conducting the measurement invariance analysis. Incorporating this in the limitations reflects their commitment to transparency and scientific rigor.
The revised discussion and improvements in language have given the manuscript greater clarity and utility, reinforcing its potential impact on the literature.
Congratulations on the work done and the adjustments achieved. Your contribution is valuable to the field.